# An abscisic acid-responsive protein interaction network for sucrose non-fermenting related kinase1 in abiotic stress response

Carina Steliana Carianopol [1,2], Aaron Lorheed Chan[1,2], Shaowei Dong[2], Nicholas J. Provart[2,3], Shelley Lumba [2] & Sonia Gazzarrini [1,2 ✉]

Yeast Snf1 (Sucrose non-fermenting1), mammalian AMPK (5′ AMP-activated protein kinase) and plant SnRK1 (Snf1-Related Kinase1) are conserved heterotrimeric kinase complexes that re-establish energy homeostasis following stress. The hormone abscisic acid (ABA) plays a crucial role in plant stress response. Activation of SnRK1 or ABA signaling results in over-lapping transcriptional changes, suggesting these stress pathways share common targets. To investigate how SnRK1 and ABA interact during stress response in *Arabidopsis thaliana*, we screened the SnRK1 complex by yeast two-hybrid against a library of proteins encoded by 258 ABA-regulated genes. Here, we identify 125 SnRK1- interacting proteins (SnIPs). Network analysis indicates that a subset of SnIPs form signaling modules in response to abiotic stress. Functional studies show the involvement of SnRK1 and select SnIPs in abiotic stress responses. This targeted study uncovers the largest set of SnRK1 interactors, which can be used to further characterize SnRK1 role in plant survival under stress.

[1] Department of Biological Sciences, University of Toronto Scarborough, 1265 Military Trail, Toronto, ON M1C 1A4, Canada. [2] Department of Cell and Systems Biology, University of Toronto, 25 Willcocks Street, Toronto, ON M5S 3B2, Canada. [3] Centre for the Analysis of Genome Evolution and Function, 25 Willcocks Street, Toronto, ON M5S 3B2, Canada. ✉email: gazzarrini@utsc.utoronto.ca

Climate change, coupled with an increase in world population, greatly limits plant productivity and yield worldwide, making it crucial to understand mechanisms of plant stress response[1]. A key player in plant stress response is the hormone abscisic acid (ABA), whose levels rise dramatically in plants exposed to environmental stress. In the presence of ABA, PYRABACTIN RESISTANCE1 (PYR1)/PYR1-like (PYL)/REGULATORY COMPONENT OF ABA RECEPTORS (RCAR) interact with, and inhibit, clade A protein phosphatase 2C (PP2C) activity. This allows SnRK2 activation and phosphorylation of downstream targets, including transcription factors and anion channels. Several basic leucine zipper (bZIP) transcription factors, such as ABSCISIC ACID5 (ABI5) and ABA-response-element binding (AREB)/ABRE-binding factors (ABF), are phosphorylated by SnRK2 kinases and bind the promoter of target genes to induce ABA-dependent gene expression[2–4]. Recently, systems biology approaches have been instrumental for the identification of novel components of the ABA signaling network and identified novel regulators of abiotic stress responses[4–6].

Regardless of the type of stress to which plants are exposed, the outcome is energy deprivation from decreased photosynthesis and respiration rates to preserve resources. SnRK1/Snf1/AMPK are conserved heterotrimeric kinase complexes that are activated under energy limitation to induce metabolic reprogramming and achieve energy homeostasis in plants, yeast, and mammals[7–9]. In humans, AMPK has been implicated in several diseases, and activation of AMPK has several metabolic and therapeutic (anti-inflammatory and anti-tumorigenic) roles[8,10]. In plants, SnRK1 signaling regulates growth, development and stress adaptation through metabolic reprogramming[7,9,11,12]. Unlike in yeast and mammals, a large SnRK superfamily has evolved in plants, which includes SnRK1, the core ABA signaling SnRK2 and the calcium-regulated CIPK/SnRK3 kinases. SnRK2/3 do not work in heterotrimeric complexes and share little sequence similarity outside the conserved kinase domain. Accordingly, SnRK2/3 cannot functionally substitute for SnRK1, suggesting functional diversification[7].

The SnRK1 heterotrimeric kinase complex comprises the α catalytic subunits (SnRK1α1/AKIN10 and SnRK1α2/AKIN11), involved in substrate phosphorylation, and three regulatory β (SnRK1β1-β3) and γ (SnRK1βγ) subunits, with complex scaffolding and subcellular localization functions[7,11,13]. In response to low energy, SnRK1 is activated by the SnRK1 activating kinase1 and 2 (SnAK1/2)/geminivirus REP interacting kinase1 and 2 (GRIK1/2), which phosphorylate SnRK1α on the conserved T-loop, similarly to the yeast and mammalian counterparts[7]. Overexpression of SnRK1α1 greatly delays germination and growth, while a double *snrk1α1 snrk1α2* RNAi/VIGS-silenced mutant is lethal, supporting the SnRK1 function in metabolic regulation and suggesting additional roles in growth and development[14]. In protoplasts overexpressing SnRK1α1, almost 1000 genes were differentially regulated, including transcription factors and other signaling components. These changes were shown to promote major catabolic pathways and repress anabolic processes[14]. Similar processes were affected in conditional *snrk1α1 snrk1α2* double mutants and shown to be regulated by SnRK1 phosphorylation targets, the S1-bZIP transcription factors[15,16]. SnRK1α can phosphorylate ABI5 in vitro and is itself dephosphorylated and inactivated by the PP2Cs, ABI1 and AHG3[17,18]. It was proposed that activation of SnRK1 signaling could be priming plant responses to stress, while SnRK1 dephosphorylation by PP2Cs may act to reset the system once optimal conditions are achieved[17].

Despite the essential role of SnRK1 in plant growth and stress response, the molecular mechanisms regulating SnRK1 complex function are largely unknown and only few substrates have been identified so far[11]. This is in contrast to 216 physical interactors identified for Snf1 in yeast[19] and over 60 targets identified through phosphorylation prediction and high-throughput studies for the mammalian AMPK[8,10], pointing to a need for large scale identification of SnRK1 interactors in plants. Activation of SnRK1 or ABA treatment results in 20–30% overlapping transcriptional changes, suggesting these important stress pathways share common target genes[17]. Here, we use a targeted high-throughput yeast two-hybrid (Y2H) approach to uncover SnRK1 interactors among a library of proteins encoded by ABA-responsive genes. The ABA-responsive SnRK1 interaction network constitutes an important resource to further study SnRK1 role in ABA and stress responses.

## Results

**The ABA-responsive SnRK1 interaction network**. To identify proteins that are regulated by ABA and SnRK1 and may play a role in plant stress responses, the six SnRK1 subunits (α1, α2, β1, β2, β3, and βγ) previously shown to form SnRK1 complexes in vivo[20–22] were screened against a library of proteins encoded by 258 ABA-regulated genes[5]. High-throughput Y2H interaction assays were conducted in duplicate using two reporters, *LacZ* and *LEU*, as previously described[5] (Fig. 1a). A total of four interaction plates (two per reporter system) were analyzed for each SnRK1 subunit (Supplementary Fig. 1). Out of 1548 possible interactions, the primary screen yielded 565 candidate pairs that tested positive in at least 1 of 4 plates (Fig. 1a and Supplementary Data 1). Interactions detected multiple times with the same reporter or with different reporters are of higher confidence[23,24], therefore, only interactions that were confirmed in duplicate, with one or both reporters, were deemed as high confidence interactions (HCI). Additionally, interactors that tested positive in all plates with all subunits were deemed unspecific and removed from further analysis, as they may represent false-positives[23]. This stringent curation of the primary screen resulted in 281 HCI, representing 18% of the interaction space (Supplementary Data 2 and Fig. 1a). More specifically, 44% tested positive with both reporters, while 56% tested positive either with the *LacZ* (41%) or the *LEU* (15%) reporter (Supplementary Data 2). Given that interactions detected with multiple reporters do not necessarily indicate interactions of higher confidence than those found with a single reporter[25], all 281 interactions were deemed to be HCI and were further analyzed. Overall, the Y2H screen yielded 125 unique SnRK1 Interacting Proteins (SnIPs) (Fig. 1a, b).

We compared our dataset with those obtained from two large-scale proteomic studies, the Y2H screen from The Arabidopsis Interactome Mapping Consortium (2011) and the split-ubiquitin screen[26]. We found no overlap with our dataset; however, related family members were identified (Supplementary Data 3). Thus, our targeted screen greatly expands the suite of SnRK1 complex interactors and provides new tools to investigate molecular mechanisms of SnRK1 function in ABA and stress responses.

**Quality of the SnRK1 interaction network**. To determine the sensitivity and background of our assay, we assembled a positive reference set (PRS) and a random reference set (RRS), respectively[25,27–29]. The PRS included all nine literature-curated interactions between SnRK1 and SnIPs that were present in the library (Supplementary Table 1). For the RRS, we selected 153 random protein pairs that did not interact in the primary screen and could be retested with both reporters (Supplementary Table 2). Assay sensitivity, determined as the fraction of PRS confirmed with each reporter, was 50 and 67% with the *LacZ* and *LEU* reporters, respectively (Fig. 1c and Supplementary Table 1). Sensitivity decreased to 50% when both reporters were scored

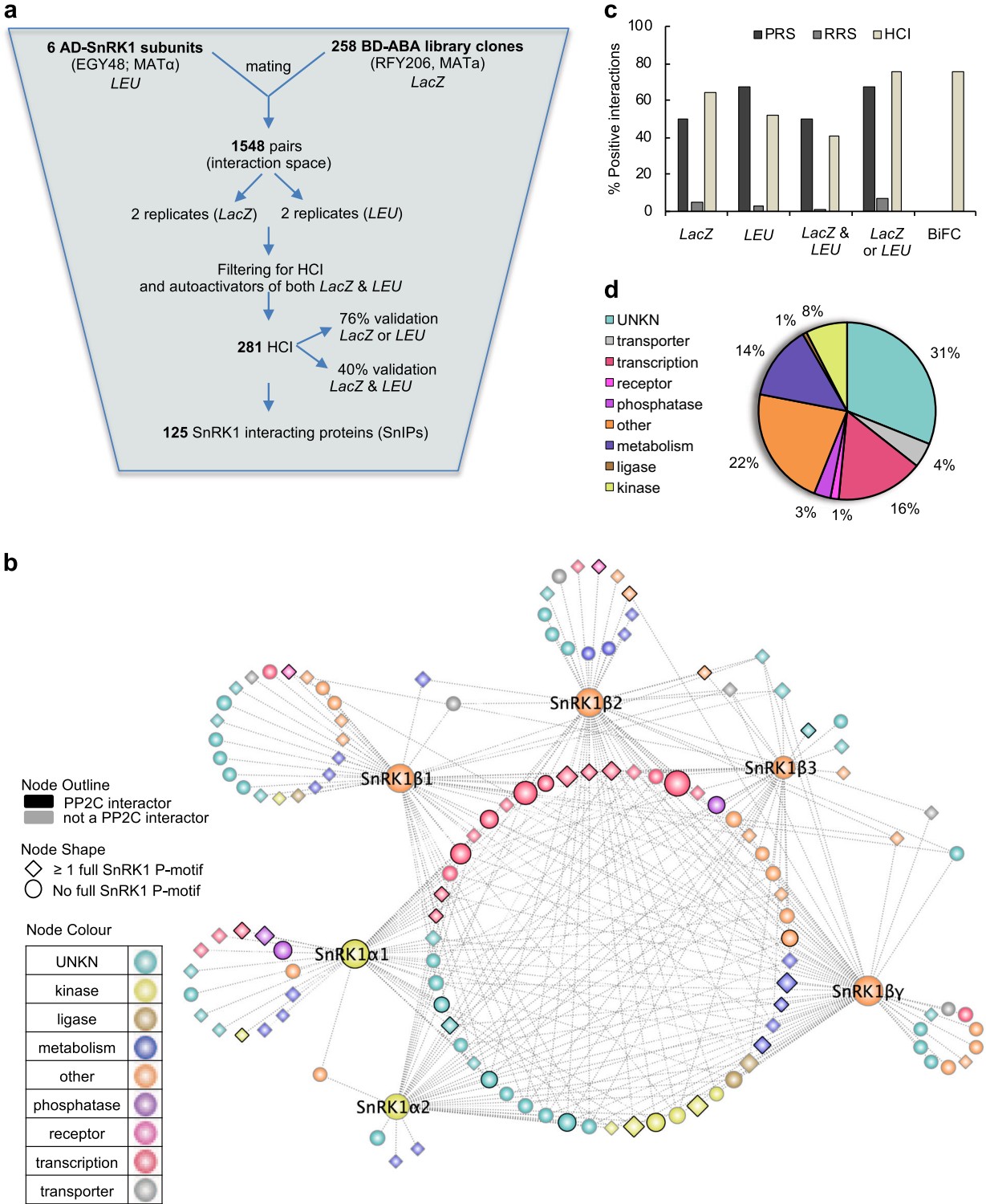

**Fig. 1 ABA-responsive SnRK1 interaction network. a** Overview of the Y2H screen. Six SnRK1 subunits were fused to the GAL4 activation domain (AD) and mated with 254 ABA library clones fused to the LexA DNA binding domain (BD). Positive interactors were scored on solid media lacking leucine or supplemented with X-gal. Two replicate plates per reporter were conducted for each of the six SnRK1 subunits. **b** Representation of the ABA-responsive SnRK1 interaction network. Forty-eight core SnRK1-interacting-proteins (SnIPs) partnered with SnRK1α and β/βγ subunits (nodes in the large central circle). Proteins (nodes) are colored based on GO molecular function. **c** Y2H assay sensitivity and background determined with a positive (PRS) and random (RRS) reference set, respectively. One hundred and thirty two high confidence interactions (HCI) were retested by Y2H and eight by BiFC. **d** GO Slim molecular function distribution of the 125 SnRK1 complex interacting proteins (SnIPs).

(positive interactions in all 4 replicates), compared to 67% when either reporter was scored (positive interactions at least in 2/4 replicates; Fig. 1c; Supplementary Table 1). Our assay sensitivity was similar to that obtained when analyzing a PRS from the Lumba et al.[5] dataset (library and screening system were identical), which was 47% (Supplementary Table 3). Assay background, determined by the fraction of interactions in the RRS that scored positive, was 4.6% with the *LacZ* reporter and 2.6% with the *LEU* reporter (Supplementary Table 2 and Fig. 1c). As expected, when scoring positive interactions resulting from either reporter, the background increased to 6.5%, while decreasing to 0.7% when scoring positive interactions resulting from both reporters. Our assay performance (sensitivity versus background) was higher than those determined in previous high-throughput Y2H screens, likely due to the library size, Y2H technique, vectors used and stringency of scoring criteria. However, a similar trend was observed, as assay sensitivity in previous studies increased from 15 to 40% depending on the number of reporters scored and vectors used at the expense of assay background[25,27,30,31].

SnRK1 complexes in vivo have been shown to include the α, β and βγ regulatory subunits, with the β subunits giving substrate specificity to the SnRK1 complex[13,20,22,32]. Of the 125 SnIPs identified, 48 were deemed core SnRK1 complex interactors (core SnIPs), as they partnered with at least one of the α catalytic subunits and one of the β/βγ regulatory subunits, increasing the confidence that these 184 SnRK1-SnIP HCI may be biologically relevant (Supplementary Data 4 and Fig. 1b). Next, we independently mated and retested the HCI with both reporters. We excluded interactions that could only be tested with one reporter, resulting in a subset of 132 HCI (47% of the total HCI). HCI were confirmed with the *LacZ* and *LEU* reporters at a rate of 64 and 52%, respectively (Fig. 1c and Supplementary Tables 4, 5). Similarly to the PRS, the confirmation rate for the HCI was higher when interactions were scored with either reporter (76%) than when scored with both reporters (40%) (Fig. 1c and Supplementary Table 4). Overall, the HCI confirmation rates were similar to, or higher than those obtained with the PRS (Fig. 1c) and are in agreement with previous studies[29]. This suggests our dataset is of high quality.

Subcellular localization predictions show that SnIPs localize to different cellular compartments (Supplementary Data 2). To evaluate the overall biological relevance of the SnRK1–SnIP interactions, we performed co-subcellular localization tests, with the rationale being that for proteins to interact they should at least some of the time be in the same subcellular compartment. For SnRK1α1, α2, β3, and βγ there was an enrichment for their interactors being in the same subcellular compartment (nucleus, based on GO Slim "Cellular Compartment" data from TAIR) as compared to expected numbers by chance, within the set of 258 library proteins (Supplementary Fig. 2a). Furthermore, Gene Ontology (GO) molecular function analysis shows that 29% of the 125 SnIPs include signaling proteins (transcription factors, phosphatases, and kinases), while 14% are involved in metabolic processes and 31% have unknown molecular function, showing no particular enrichment for any category compared to the library tested (Fig. 1d). This is in agreement with SnRK1 well-established role in metabolism and signaling. Co-molecular function enrichment analysis (using GO Slim "Molecular Function" data from TAIR) showed that SnRK1β2 and βγ exhibited co-molecular function enrichment ("protein binding") for their interactors as compared to expected within the library tested (Supplementary Fig. 2b). Both of these tests partially support our experimental results, whereby we acknowledge that two interacting proteins do not necessarily have to have the same molecular function and that the winner-takes-all approach for subcellular localization co-enrichment analysis would result in

false negatives in the test. Altogether, these results suggest that the 281 HCI have a high probability of occurring in planta.

**Enrichment of potential SnRK1 kinase substrates in the ABA-regulated library.** The 125 SnIPs include known SnRK1 subunit modulators (AHG3), scaffolding proteins (DUF581-12[33,34]) and kinase substrates (ABI5, ABF3). To determine the fraction of potential SnRK1 substrates, we created PhosMoS (Phosphorylation Motifs of SnRK1), a Python script that identifies the SnRK1 consensus sequence ([MLVFI]-[XRKH]-[XRKH]-XX-[ST]-XXX-[LFIMV][35]). Our analysis shows that 56% of the 125 SnIPs and 46% of the 48 core SnIPs contain a full SnRK1 motif (Supplementary Data 4 and Fig. 1b); these include four known interactors (ABI5, ABF3, SnRK3.15, and ANAC002), including phosphorylation substrates (ABI5, ABF3). Notably, the 258 ABA-responsive library proteins are enriched for proteins with a full SnRK1 motif compared to the whole Arabidopsis proteome (58% in the library vs. 30% in the whole proteome; Fisher's exact test, $p < 0.001$). Protein phosphorylation is a reversible post-translational modification, which relies on the action of protein kinases and phosphatases. The SnRK1 kinase substrates, ABF3 and ABI5, have been shown to interact with clade A PP2C phosphatases, suggesting PP2Cs can directly regulate not only SnRKs but also their substrates[36]. Interestingly, the 48 core SnIPs show enrichment for PP2C interactors, compared to the 125 SnIP set or the ABA library (21 of the 48 core SnIPs, compared with 40 of the 258 ABA library; Fisher's exact test, $p = 0.0002$), suggesting that some SnIPs may be phosphorylated by SnRK1 kinases and dephosphorylated by PP2C phosphatases (Supplementary Data 4 and Fig. 1b). Taken together, this analysis suggests that SnRK1 may play a prominent role in modulating the ABA response pathway.

**SnRK1 interacts with ABA response and stress signaling pathways.** GO biological process enrichment of the 125 SnIPs, compared to the Arabidopsis genome, shows that the SnRK1 network is enriched for proteins involved in ABA signaling ($p = 1.5 \times 10^{-8}$), regulation of metabolic processes ($p = 0.036$), sugar signaling ($p = 0.046$) and salt stress response ($p = 0.017$) (Supplementary Data 5). This is in agreement with the well-known role of SnRK1 as regulator of metabolism and its function in transcriptional reprogramming following stress[11,14,17]. Enrichment in ABA signaling is expected, given that the library consisted of ABA regulated proteins. We next analyzed and compared the suites of interactors of the SnRK1α catalytic and SnRK1β/βγ regulatory interactors.

The SnRK1α1 and α2 catalytic subunits interacted with 68 highly interconnected proteins, representing 54% of total SnIPs, 25 of which were shared between the two α subunits (Supplementary Fig. 3a, b; Supplementary Data 2). The majority of SnRK1α interactors are signaling proteins (41%), while 15% are involved in metabolism and several have unknown function. SnRK1α shared interactors include both positive (ABI5) and negative (AFP2, HB6, and AIB1) regulators of the ABA signaling pathway[4,5,37], as well as transcription factors and kinases involved in ABA signaling and abiotic stress response (ANACs and DIG1/AITR2[6,38,39]). SnRK1α1-specific partners include several ANACs, while FER1, involved in nutrient starvation response[40], was among the SnRK1α2-specific interactors. Differences in SnRK1α1 and α2 partners may be due to their divergent C-terminal protein-protein interaction domain (Supplementary Fig. 3c, d). This suggests that the two catalytic subunits may play different as well as redundant roles in ABA and stress responses.

The regulatory SnRK1β/βγ subunits, which have divergent protein domain structures, partnered with 132 proteins

(Supplementary Data 2). The 57 SnRK1β/βγ-specific interactors (not interacting with α subunits) comprise proteins involved in ABA metabolism (CYP707A1), perception (PYR1 and PYL4) and nutrient transport (Pht1;4 phosphate transporter, SUC1, and STP4; Supplementary Fig. 4a, c, d). However, the majority (39%) of SnRK1β/βγ-specific interactors have unknown molecular function and many proteins do not partner with other interactors within the SnRK1 network (Supplementary Fig. 4a, b).

Genetic analysis shows that the *snrk1a1* and *snrk1a2* knockout mutants are hypersensitive to 0.5–1 μM ABA, suggesting that at these concentrations SnRK1α attenuates the ABA response (Supplementary Fig. 5a, b). Taken together, these data show that SnRK1 interacts with the ABA pathway at multiple levels, including metabolism, perception, and response, and thus may regulate ABA-mediated stress responses (Supplementary Fig. 5c).

**SnRK1 signaling modules during abiotic stress**. Comparisons with external transcriptomic and proteomic data have shown that proteins that are co-regulated in expression and highly interconnected (share similar interaction partners) are more likely to constitute a biologically relevant signaling module and/or to belong to the same protein complexes[41,42]. Given that SnRK1 is a heterotrimeric complex, we sought to determine whether the 48 core SnIPs that partner with at least one of the SnRK1α and one of the SnRK1β/βγ subunits (Fig. 1b and Supplementary Data 4), form signaling modules during abiotic stress.

We first generated a force-directed representation of the 48 core SnIP subnetwork, which clusters SnIPs based on interactor commonality (Fig. 2a). To this end, we retrieved all previously published interactions with the 48 core SnIPs from the Arabidopsis Interaction Viewer and found that the 48 core SnIPs form a highly interconnected network comprising 314 total interactions (204 SnRK1–SnIP and 110 SnIP–SnIP). Of note, 45% of the 48 SnIPs are involved in signaling, with enrichment for transcription factors (28% vs. 14% in ABA library; Fisher's exact test, $p = 0.0174$) (Fig. 2b). Next, we found that the 48 core SnIPs are co-expressed and mostly upregulated during abiotic stress (Fig. 3a). Although the SnRK1 subunits are ubiquitously expressed and regulated primarily at the post-translational level[43], we found that SnRK1 subunits are differentially regulated under prolonged exposure to osmotic/salt stress in shoots (Fig. 3a). Lastly, we identified tissue-specific interacting proteins that are significantly correlated in expression under stress (Pearson Correlation Coefficient, PCC cut-off of 0.75; $p < 0.01$). We found that 19 (6%) of the interactions are statistically correlated under osmotic stress in roots, 33 (11%) under osmotic stress in shoots, 37 (12%) under salt stress in roots and 37 (12%) under salt stress in shoots (Fig. 3b and Supplementary Data 6). Altogether, this shows that the 48 core SnIPs are enriched for transcription factors, and are highly interconnected and co-expressed under abiotic stress, suggesting they may form signaling modules in different tissues and under different stresses.

To identify highly-connected and tissue-specific interacting proteins during stress, we constructed edge-weighted force-directed SnRK1-SnIPs interaction networks (Fig. 4). We focused our analysis in the shoot, where SnRK1 expression level changes under prolonged exposure to osmotic and salt stress (Fig. 3). Proteins that are upregulated in expression under stress and also share a higher number of co-expressed interacting partners (cluster together) are more likely to interact and represent stress modules (Fig. 4). Using this criteria, 29 central abiotic stress SnIPs were identified under salt and/or osmotic stress (Supplementary Data 4). Under both osmotic and salt stress, similar SnIPs can be observed such as ABI5, ABF3, SNRK3.15, SNRK3.22, AFP2, DUF828/FL7, AIB1, and

DIG1, while others such as MYB77 appear to be salt stress-specific. Supporting our prediction, five of the 29 central abiotic stress SnIPs have been shown to play a role in abiotic stress: DIG1[6] and MAP3Kδ4[44] in salt stress, while AFP2[45], ABI5, and ABF3[46] in both osmotic and salt stress. This suggests that the interactions in the SnRK1–SnIP networks are dynamic, and different protein-protein interactions may be important under specific stress conditions, while others may be general stress responsive pairs.

**Validation of core SnRK1–SnIP interactions**. To validate the HCI identified by Y2H, we confirmed selected SnRK1α interactions in planta by Bimolecular Fluorescence Complementation (BiFC). BiFC and Y2H are orthogonal assays used to validate results obtained with either technique[23]. Furthermore, BiFC allows the visualization of the subcellular location of the interaction, in addition to detecting transient kinase-substrate interactions, which are facilitated by the reconstitution of the stable, full-length yellow florescent protein (YFP)[28,47]. We selected interactions between SnRK1α1/α2 and representative central abiotic stress SnIPs that are involved in signaling, or have unknown function, and also have a putative full SnRK1 phosphorylation site (Supplementary Data 4). This filtering resulted in 12 candidates, three of which are known SnRK1α interacting proteins (ABI5, ABF3, and SnRK3.15). We validated the following five candidates, for a total of ten interactions: SnRK3.22; MYB77; the less characterized DUF828/FORKED-LIKE7 (FL7), which has three predicted SnRK1 motifs (Supplementary Data 4); AFP2, which is known to play a role in abiotic stress but does not have a full SnRK1 motif and may be a complex modulator/adapter; and ANAC055, which was predicted to be a false positive by our filtering criteria. The MYB77-SnRK1α2 pair did not interact in the primary screen, and thus served as negative interaction. We first confirmed their expression in planta by fusing SnIPs to full-length YFP (Fig. 5a). Then, we tested interactions between each pair of proteins by fusing SnRK1 and SnIPs to both the N-terminal half and C-terminal half of YFP (Fig. 5b).

Six interactions were validated by BiFC: (i) SnRK1α1 with SnRK3.22, AFP2, DUF828, MYB77, and (ii) SnRK1α2 with SnRK3.22, AFP2. SnRK1α2 and MYB77 did not interact in planta (Fig. 5b), confirming the Y2H results and suggesting that MYB77 is indeed an α1 specific interactor. DUF828/FL7 and SnRK1α2 interaction was not validated using this method, despite being confirmed in the HCI retest. ANAC055 tested negative using BiFC with both SnRK1α1 and α2, confirming that it is likely to be a false positive and supporting our filtering criteria. All BiFC interactions were specific, as no fluorescence was detected with the vector controls harboring the half-YFP (Supplementary Fig. 6). Overall, six of the eight (75%) positive Y2H interactions were validated in planta, a rate similar to those obtained for PRS and HCI validation assays (Fig. 1c). This is in agreement with previous high-throughput verifications rates using an independent assay[5,48] and confirms that our curated Y2H interactions are of high quality. Additionally, these interactions occur in the nucleus, consistent with recent studies which have shown that SnRK1 relocates to the nucleus in response to stress[13,49].

**Role of SnRK1 and core SnIPs during stress response**. To investigate if SnRK1α and the in planta validated SnIPs play a biological role under osmotic or salt stress in vivo, we tested the sensitivity of *snrk1a1, snrk1a2, snrk3.22, duf828/fl7, myb77* and *afp1-4 afp2-1* mutants to these stresses during early seedling growth. All mutants showed 100% cotyledon expansion on control media, however under high osmotic stress (400 mM sorbitol), only *snrk1a2* and *afp1-4 afp2-1* showed reduced cotyledon

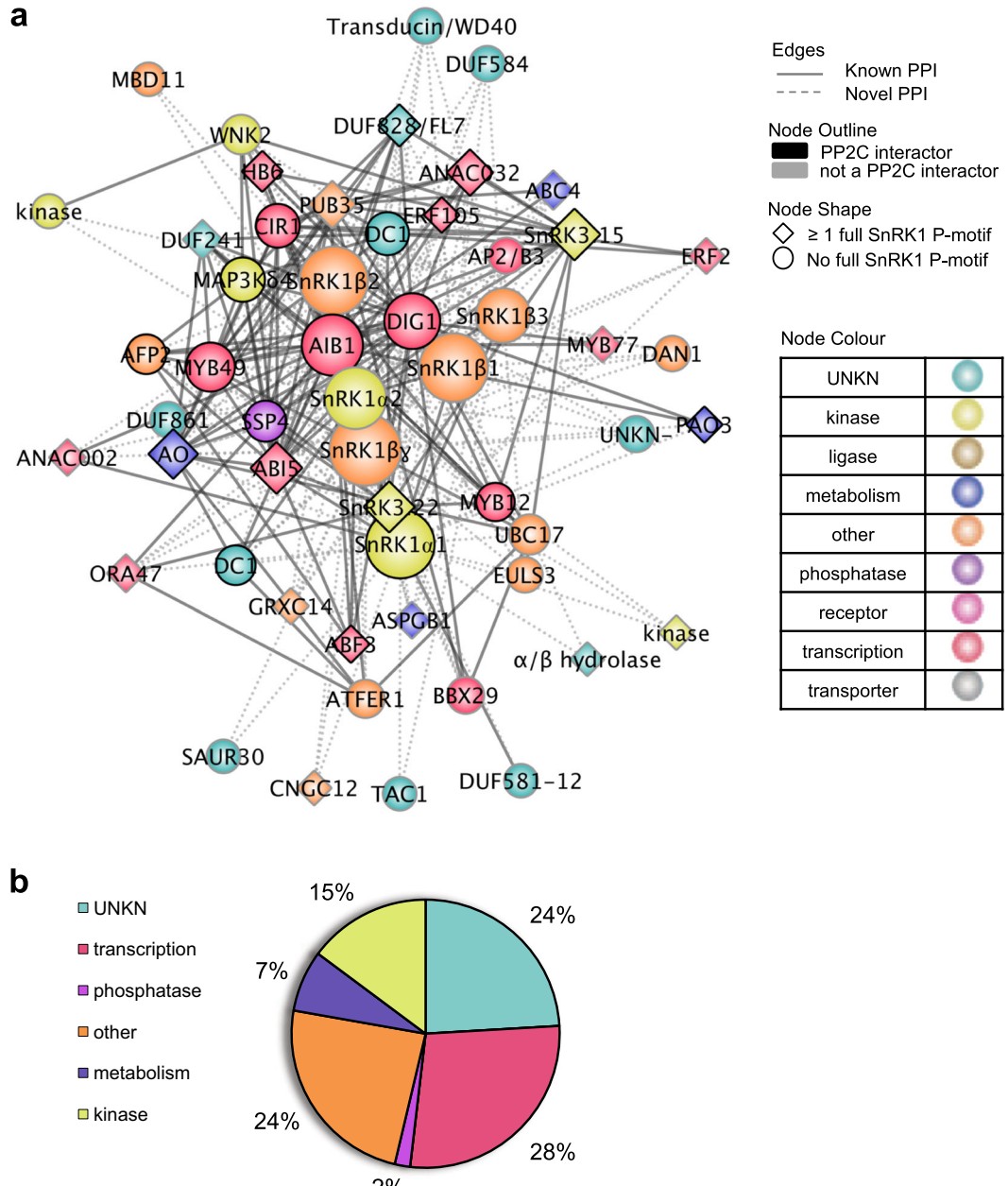

**Fig. 2 Extended interaction network of 48 core SnIPs. a** Force directed representation of the 48 core SnRK1 complex interacting proteins (SnIPs). Proteins (nodes) are colored based on GO molecular function. Solid lines denote previously published protein–protein interactions (PPIs) obtained from the Arabidopsis interaction viewer. Dotted lines indicate PPI identified in this study. Black node outline shows interaction with any PP2C in the ABA-responsive library[5]. Node shape indicates whether proteins contain a full SnRK1 phosphorylation site. **b** GO Slim molecular function distribution of the 48 core SnIPs.

expansion compared to wildtype (ANOVA $p = 5.19 \times 10^{-13}$ post hoc Tukey HSD test $p = 1.7 \times 10^{-7}$ and $p = 1.1 \times 10^{-9}$, respectively), suggesting that they are more sensitive to osmotic stress (Fig. 6a, b and Supplementary Data 7). Under moderate salt stress conditions (100 mM NaCl), the *snrk1α2, afp1–4 afp2-1*, and *snrk3.22* mutants showed hypersensitivity (reduced growth) to salinity stress compared to wildtype (ANOVA $p = 2.46 \times 10^{-16}$ post hoc Tukey HSD test, $p = 8 \times 10^{-7}$; $p = 1.2 \times 10^{-6}$; $p = 0.027$, respectively), suggesting that these genes inhibit the salt stress response (Fig. 6c, d and Supplementary Data 7). These results are consistent with AFPs and SnRK3.22 roles as negative regulators of ABA signaling[4]. In contrast, *duf828/fl7* and *myb77* (ANOVA $p = 2.46 \times 10^{-16}$, post hoc Tukey HSD test $p = 0.012$ and $p = 6 \times 10^{-7}$, respectively) showed reduced sensitivity compared to

wildtype, as shown by increased growth under salt, suggesting they function to promote the salt stress response (Fig. 6c and Supplementary Data 7). None of the mutants showed a growth phenotype different from wildtype on control media (ANOVA $p = 0.298$, 95% CI), nor when exposed to the corresponding osmotic stress (ANOVA $p = 0.115$, 95% CI).

These data show that SnRK1α2 and all novel SnIPs tested (DUF828/FL7, MYB77 and SnRK3.22) play a role in plant response to osmotic and/or salt stress during seedling establishment/growth. Our results (Fig. 6) and a survey of the literature (Supplementary Data 4) show that mutants of 11 out of the predicted 29 central abiotic stress SnIPs (representing 100% of those tested so far) display osmotic/salt stress phenotypes, supporting the bioinformatic predictions.

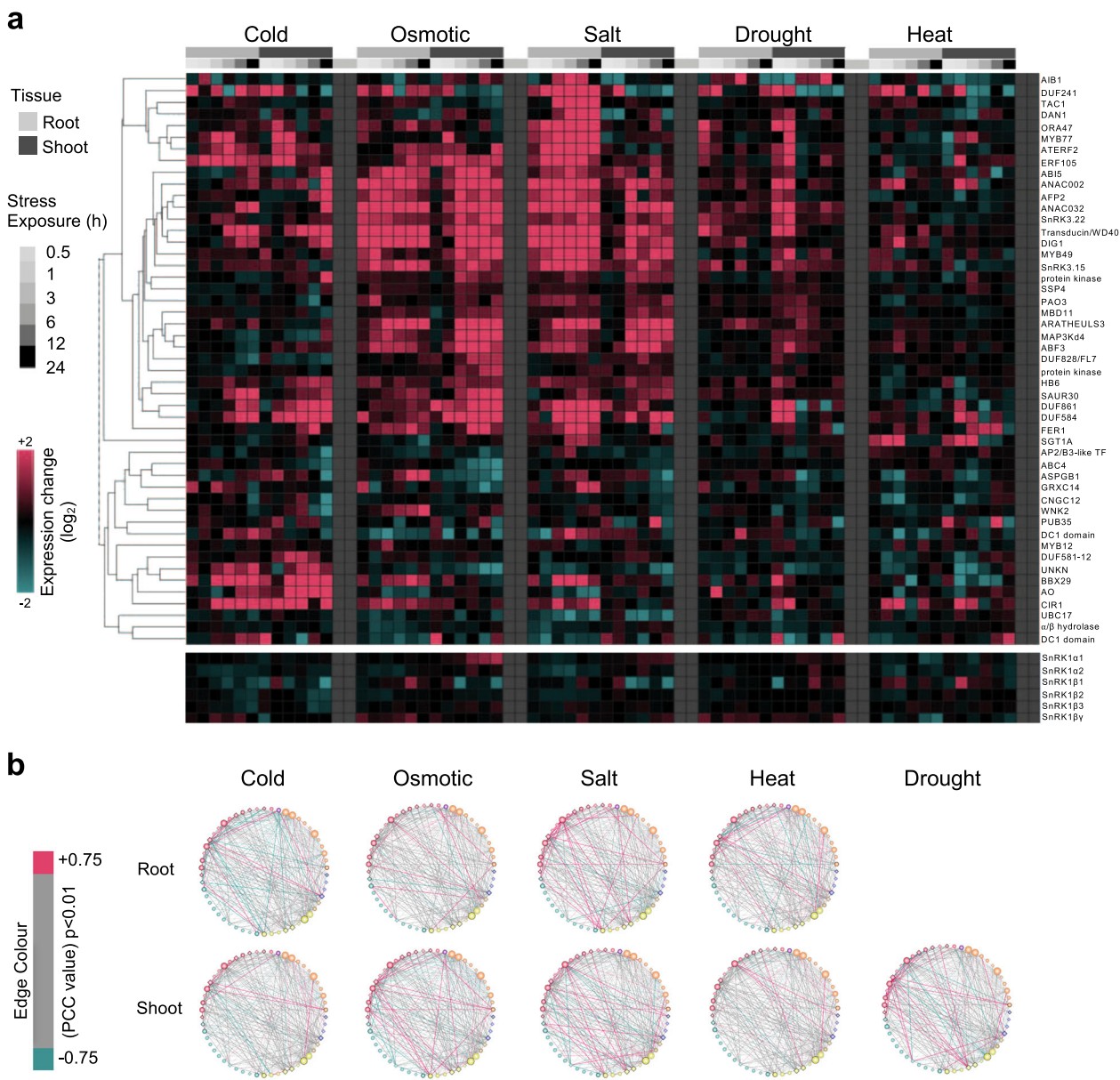

**Fig. 3 Expression correlation of 48 core SnIPs during abiotic stress. a** Heatmap representation of gene expression levels of SnRK1 and the 48 core SnIPs under abiotic stress in roots and shoots generated with expression data from AtGeneExpress (BAR; Toufighi et al.[71]; Kilian et al.[70]). **b** Co-expression map of genes encoding the SnRK1 subunits and 48 core SnIPs; each edge (protein-protein interaction) is colored based on the Pearson Correlation Coefficient (PCC; *p* < 0.01). Blue and pink edges (lines) refer to statistically significant anticorrelation (PCC < −0.75) and correlation (PCC > 0.75) in expression change, respectively.

**The SnRK1/SnRK3 kinase interaction network**. Our functional analysis uncovered a positive role for SnRK1α2 and SnRK3.22/CIPK11 kinases in the salt stress response. SnRK kinases recognize a common core phosphorylation motif [(R/K)-XX-(S/T)][35,50,51]. Thus, these kinases may share a common set of target proteins, explaining the similar phenotype observed under osmotic/salt stress. Accordingly, ABI5 is phosphorylated by SnRK1/2/3 kinases[18,52,53]. To test this, we identified proteins present in the ABA-responsive library that interact with both SnRK1 and SnRK3s, by overlaying the 125 SnIPs with the 33 interactors of SnRK3.14, SnRK3.15/CIPK14, and SnRK3.22/CIPK11 previously identified in Lumba et al.[5] (Fig. 7a). We found that 20 SnIPs interact also with SnRK3s (Supplementary Table 5). The 20 SnRK1/3 shared interactors consist of mostly proteins involved in core ABA signaling (ABI5, ABF3, HAI1, and AHG3) and ABA/abiotic stress response (ANACs, DIG1/AITR, HB6, and

AIB1)[5,6,37–39], suggesting that select transcription factors may act as a point of intersection between SnRK1 and SnRK3 pathways, possibly integrating different upstream signals and strengthening network robustness under stress.

To further understand the degree of connectivity between SnRKs, we integrated the SnRK1 interaction data collected in this screen with the SnRK3 interaction dataset from Lumba et al.[5], to cluster the 125 SnRK1 and 33 SnRK3 interactors in neighborhoods based on partner commonality (number of shared interaction partners; Fig. 7b and Supplementary Data 8). We found that SnRK1α2, SnRK3.22, and SnRK3.15 cluster in the same neighborhood, as they share several highly interconnected partners, including several ABA signaling proteins (Supplementary Table 5). In contrast, SnRK1α1 clusters in a separate neighborhood with SnRK3.14, a protein which promotes salt tolerance and regulates auxin transport[54] (Fig. 7b). This suggests

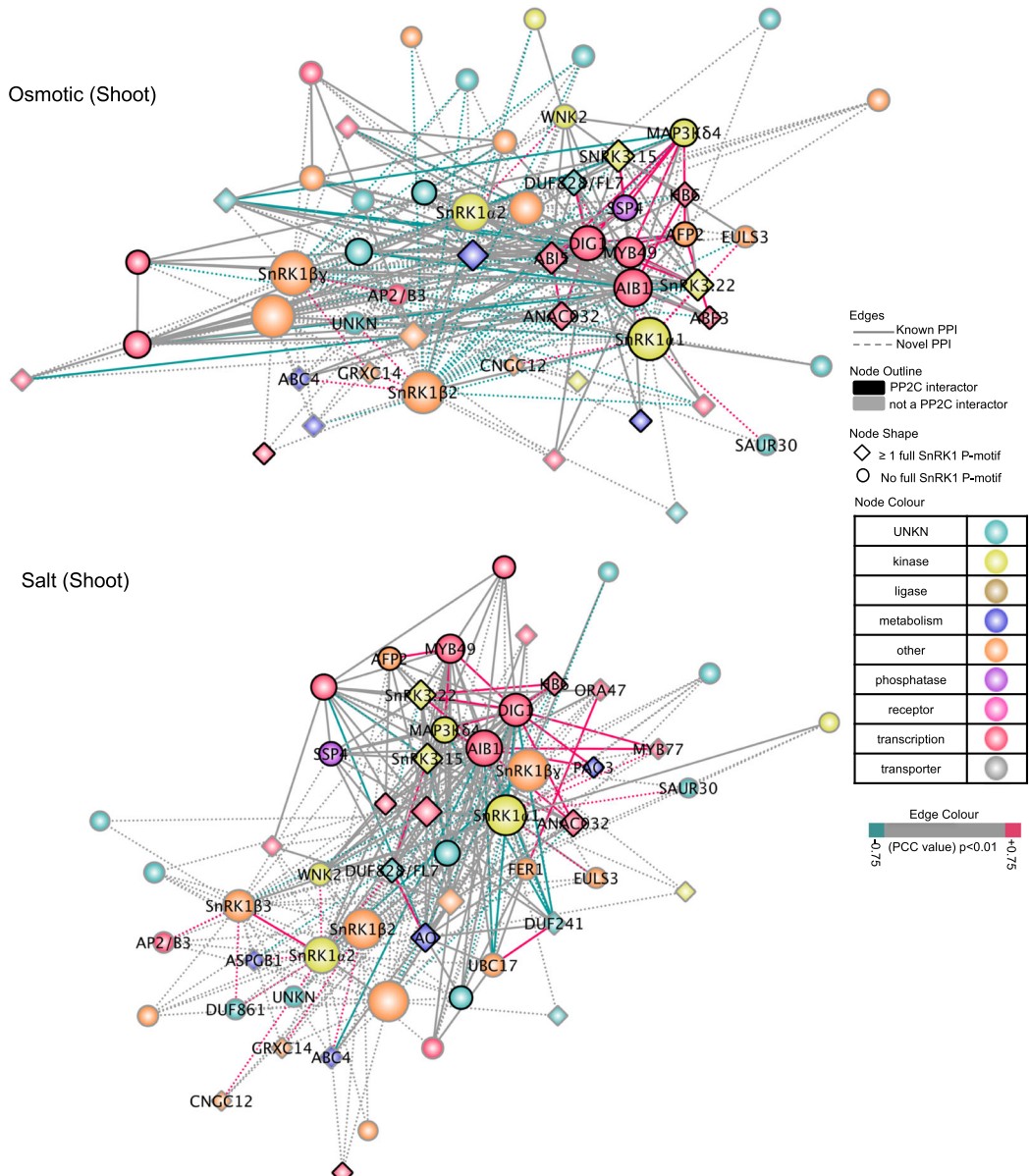

**Fig. 4 SnRK1-SnIP sub-networks under osmotic and salt stress in the shoot.** Edge-weighted force-directed interaction networks of SnRK1 and the 48 core SnIPs, under osmotic (top) and salt (bottom) stress in the shoot. Nodes closer together share a higher number of interacting partners (cluster together). Pink and blue edges (lines) connect genes (nodes) correlated and anti-correlated in expression change, respectively. Nodes which have partners that are correlated in expression are labeled and deemed as core interactors of the stress sub-network (see Supplementary Data 4).

that perturbation of SnRK function across clusters would result in stronger phenotypes than perturbations of SnRK function within the same cluster, as SnRKs within the same cluster may function in the same pathway. Accordingly, simultaneous removal of SnRK1α2 and SnRK3.22, both of which display similar phenotypes under salinity stress (Fig. 6c), does not result in enhancement of the phenotype, suggesting that these two genes work in the same pathway (Fig. 8 and Supplementary Data 7). Being interaction partners, they may modulate each other's function aside from regulating common targets. As expected, we were unable to isolate a *snrk1α1 snrk1α2* double homozygous mutant, which is lethal, as previously shown using knockout in combination with partial knockdown mutants[14,15]. This is in agreement with the network analysis and SnRK1 pivotal role in metabolism, development and stress responses. Taken together, our results suggest that our in silico network analysis is biologically relevant, and that the ABA-responsive SnRK1

interaction network can be used as a resource for hypothesis generation.

## Discussion

Environmental stress greatly affects plant productivity and poses a threat to food security. Abiotic stress such as osmotic, salt and drought leads to the accumulation of ABA, which promotes stress adaptation through the activation of stress-specific as well as shared signaling pathways[55]. Under stress conditions, plants respond by inhibiting growth to conserve energy and maximize survival. The SnRK1/Snf1/AMPK kinase complexes play conserved and essential roles in growth, development and stress responses in eukaryotes. Despite SnRK1 pivotal role in transcriptional and metabolic reprogramming following stress, only few substrates and complex regulators have been identified in plants[9,11]. This is in contrast to the network of Snf1 and AMPK substrates identified in yeast and

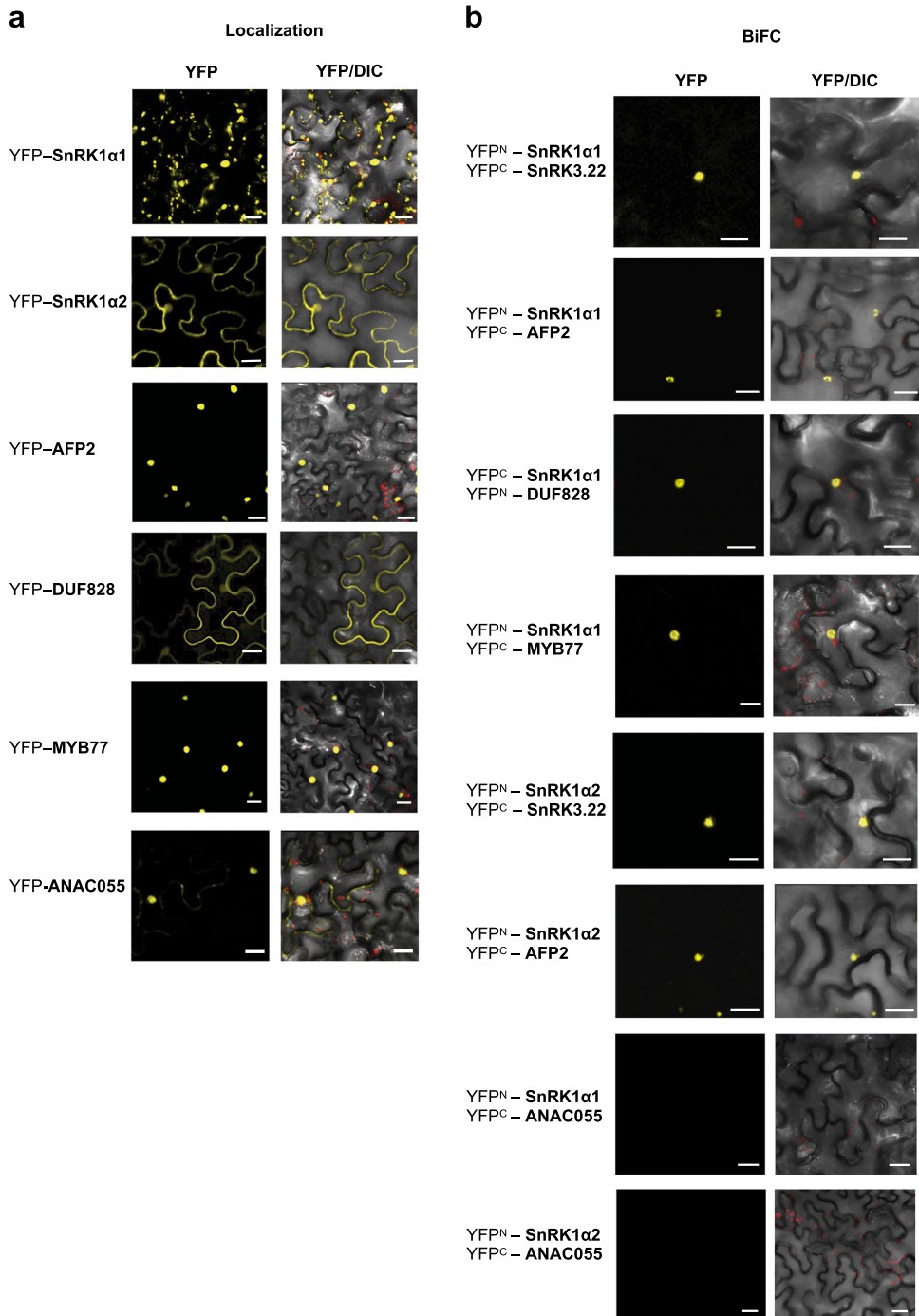

**Fig. 5 In planta confirmation of SnRK1-SnIP interactions. a** Localization of SnRK1α1/α2 and their interacting proteins fused to full-length yellow fluorescent protein (YFP). **b** Bimolecular fluorescence complementation (BiFC) assays between SnRK1α1/α2 and their interacting proteins, fused to the N or C-terminus of YFP (n/cYFP). Negative controls are shown in Supplementary Fig. 6.

mammals[8,56]. In this study, to the extent of our knowledge, 119 novel and 6 known SnRK1-complex interactors were uncovered among 258 ABA-responsive proteins, greatly expanding the suite of SnRK1-complex partners. The SnRK1 subunits interacted with core ABA signaling and response proteins, including positive and negative regulators, suggesting that SnRK1 may modulate the ABA response pathway at multiple levels. Interestingly, *snrk1α1* and *α2* were hypersensitive to ABA during seedling establishment, similarly to the phenotype exhibited by plants that overexpress SnRK1[57], suggesting that SnRK1α levels must be tightly regulated for a correct ABA response. In silico

analysis of the 125 ABA-regulated SnIPs identified a core set 48 proteins that are highly interconnected and co-expressed under osmotic and salinity stress. We showed that three SnIPs play a role in osmotic/salt stress responses and uncovered a more prominent role for the less studied catalytic subunit, SnRK1α2, in osmotic and salinity stress. A function for SnRK1 in salt stress parallels findings in yeast, where the salt stress response is dependent on Snf1[58]. Through high confirmation rates of SnRK1-SnIP interactions *in planta*, and biological roles for select SnIPs and SnRK1 in osmotic and salt stress response, the ABA-responsive SnRK1 interaction network identified here

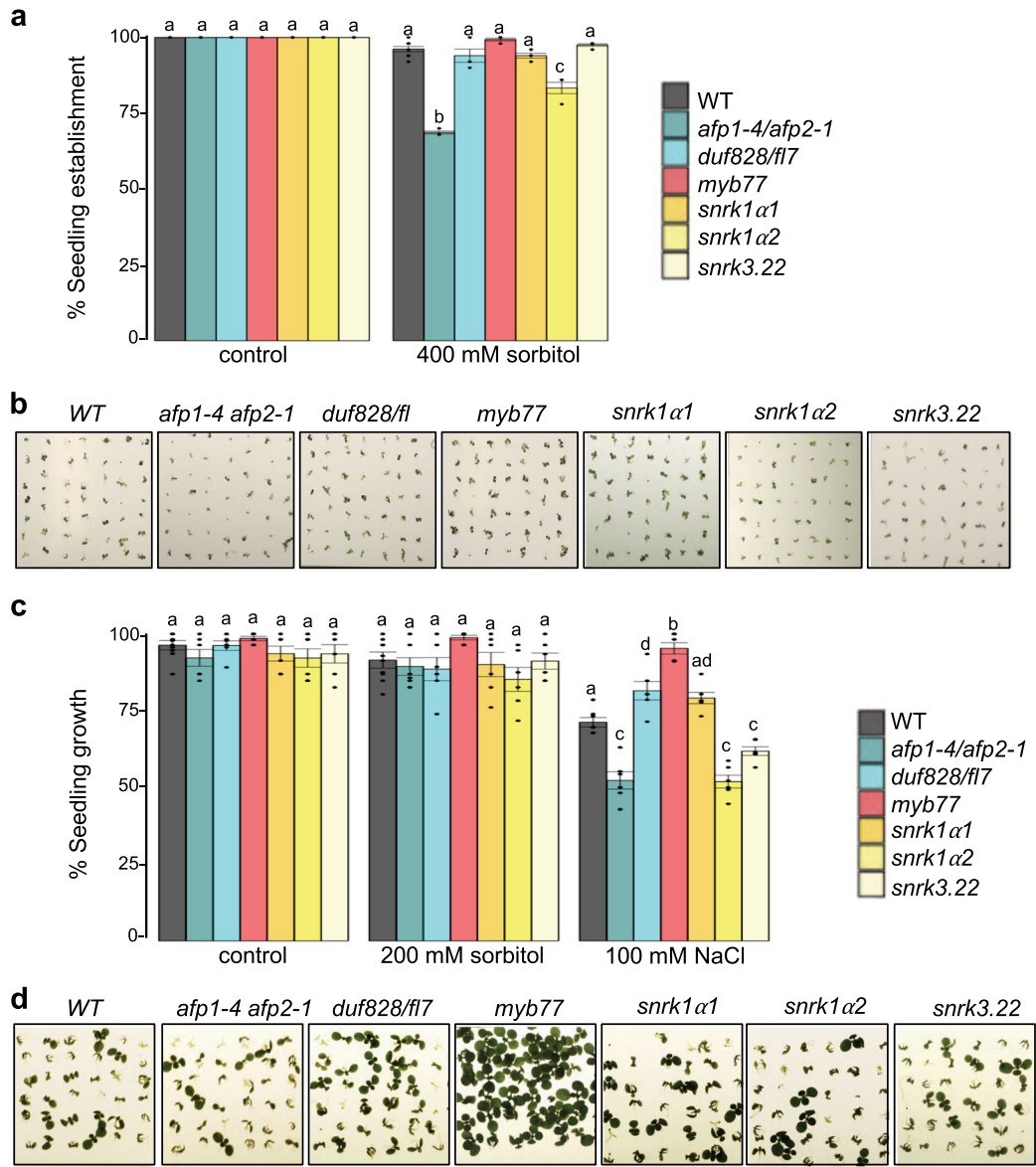

**Fig. 6 Functional analysis of SnRK1 and SnIPs under osmotic and salinity stress. a** Seedling establishment (cotyledon expansion) of *snrk1α* and SnIP mutants 10 days after imbibition (DAI). Mean of three independent biological replicates ($n = 3$) ± standard error of the mean (SEM), with 50 seedlings per biological sample. Statistical significance is shown in comparison to wildtype Col-0 within each treatment (ANOVA $p < 0.01$, post hoc Tukey HSD test $p < 0.001$). **b** Representative images of seedling establishment on 400 mM sorbitol 10 DAI. **c** Seedling growth (development of true leaves) 10 DAI. Mean of six independent biological replicates ($n = 6$) ± SEM, with 46–52 seedlings per biological sample (Supplementary Data 7). Statistical significance (ANOVA $p < 0.01$, post hoc Tukey HSD test $p < 0.05$) is shown in comparison to Col-0 wildtype (WT), within each treatment. **d** Representative images of seedling growth on 100 mM NaCl.

can be used to further investigate the role of SnRK1 in ABA and abiotic stress responses.

In this study, *snrk3.22* and *afp1-4 afp2-1* showed hypersensitivity (reduced seedling growth) to salt stress, similarly to *snrk1α2*, suggesting they function to attenuate the salt stress response. In contrast, DUF828/FL7 and MYB77 were found to have the opposite role. To our surprise SnRK1α1 did not have a prominent role in osmotic/stress response during early seedling growth. Perhaps this is due to compensatory mechanisms, leading to increased SnRK1α2 level/activity when SnRK1α1 is absent, which has been previously shown under prolonged darkness conditions in the *snrk1α1* mutant[15]. Thus, we uncovered a role for SnRK1α2, DUF828/FL7 and MYB77, in osmotic/salt stress.

DUF828/FL7 contains a pleckstrin homology (PH)-like domain, which plays a role in membrane targeting and protein–protein

interactions[59]. In Arabidopsis, DUF828/FL7 belongs to a family of eight proteins involved in vesicular trafficking and asymmetric localization of the auxin efflux carrier, PIN1[60]. Our bioinformatic and genetic analyses show that DUF828/FL7 plays a role during growth on salt. Among the mutants tested, *myb77* showed the strongest salt insensitivity. MYB77 plays a positive role in activating the auxin response pathway[61,62]. These findings point to a possible connection between SnRK1 and auxin in salt stress response.

Recently, implication of SnRK1 in salt stress response has been indirectly shown. Firstly, salt induction of the SnRK1 phosphorylation target, bZIP1, which is involved in the transcriptional reprogramming during salinity stress, is reduced in *snrkα1*[63]. Secondly, the SnRK1 upstream activating kinases, SnAKs, play a positive role in survival under salt stress, although a direct link between SnAKs and SnRK1 on salt stress was not shown[64]. Our

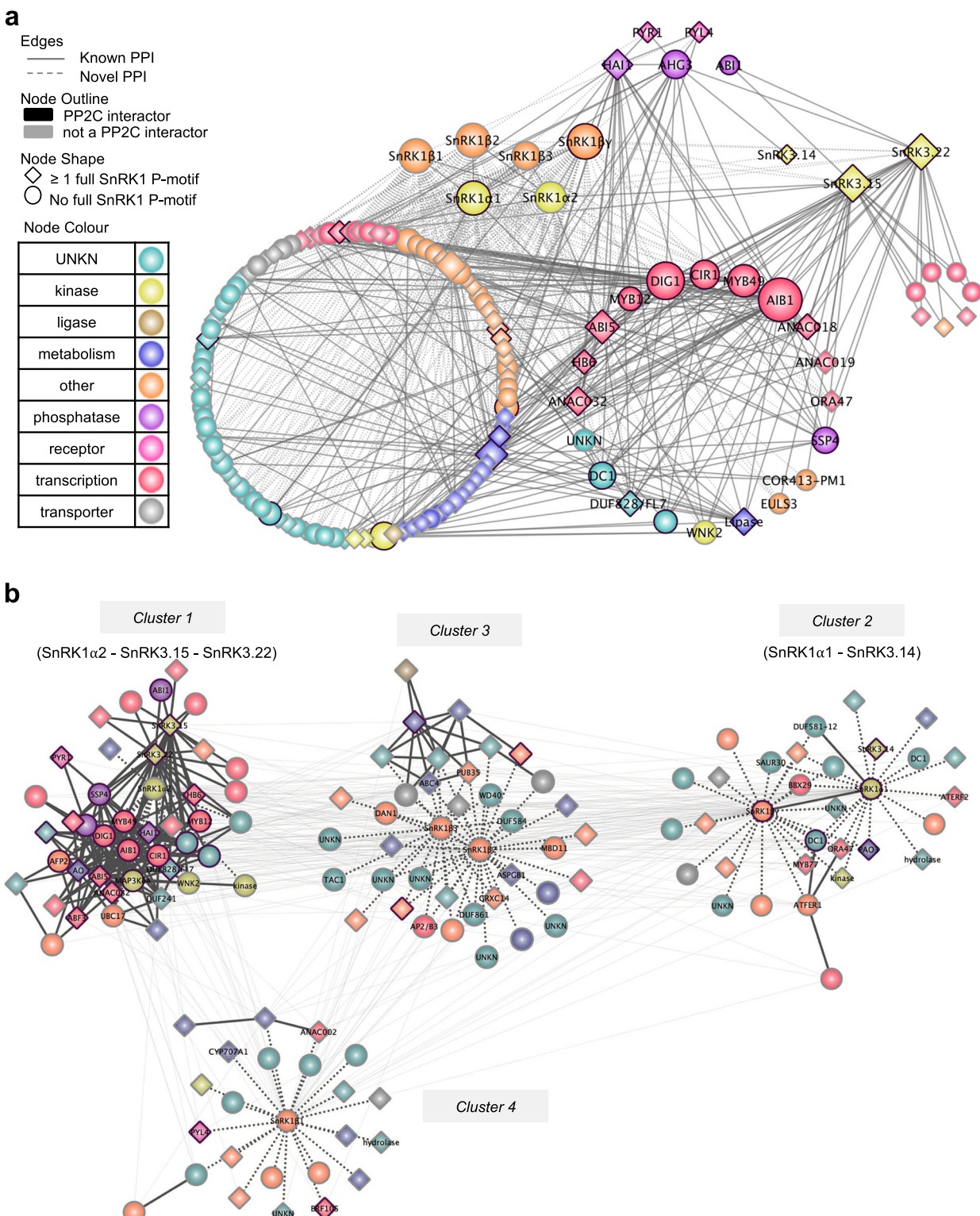

**Fig. 7 SnRK1-SnRK3 interaction network. a** Interaction network between the 125 SnRK1 interacting proteins (SnIPs) identified in this screen and the 33 SnRK3 interacting proteins identified in Lumba et al.[5]. A subset of 21 proteins interacts with both SnRK1 and SnRK3 (3.14, 3.15, and 3.22) kinases. **b** Clustering of all SnRK1 and SnRK3 interactors based on interactor commonality (interaction data from this study and[5]) using ClustalMaker plug-in. Proteins (nodes) clustering closer share higher numbers of interactors. Nodes representing the SnRK1 complex subunits, 48 core SnIPs and core ABA signaling/synthesis proteins are labeled. Nodes are colored based on GO molecular function. Dark node outline shows interaction with any PP2C in the ABA-responsive library[5]. Node shape indicates whether proteins contain a full SnRK1 phosphorylation site. Edges (gray lines) represent interactions between nodes and appear lighter when occurring between clusters, while darker within each cluster. Dotted edges (lines) indicate interactions identified in this screen; solid lines denote previously published protein-protein interactions (Arabidopsis Interaction Viewer).

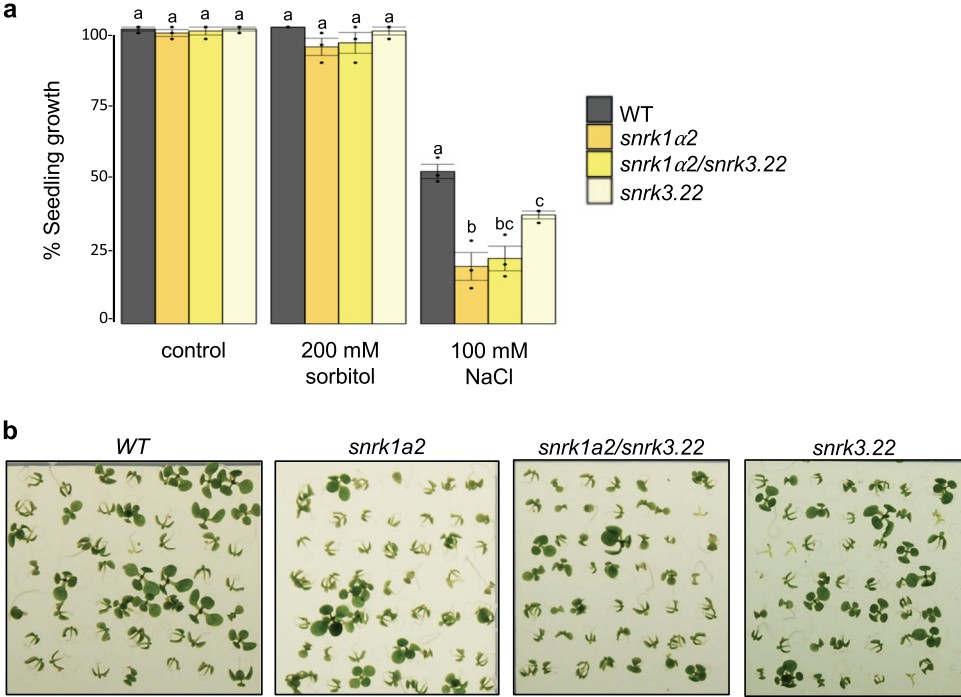

**Fig. 8 Functional analysis of *snrk1a2 snrk3.22* seedlings under salinity stress. a** Seedling growth (development of true leaves) at 10 days after imbibition (DAI) on 200 mM sorbitol and 100 mM NaCl. Means of three independent biological replicates (*n* = 3) ± standard error of the mean are shown, with 50 seedlings per biological sample. Statistical significance was calculated using ANOVA ($p < 0.05$) and post hoc Tukey HSD test ($p < 0.05$) and is shown in comparison to Col-0 wildtype (WT) within each treatment (Supplementary Data 7). **b** Representative images of seedling establishment under 100 mM NaCl stress of one biological replica.

in silico and genetic analyses support a role for SnRK1/3 and SnIPs in salt stress response. Given the role of ABA in promoting stress responses and considering the overlap in SnRK1 and SnRK3 targets among the ABA-regulated proteins found in this study, we suggest that SnRK1 and SnRK3 could phosphorylate common target proteins to promote adaptive response during salinity stress. SnRK1 also interacts with SnRK3.15 and SnRK3.22, thus these kinases may also regulate each other's activity (Supplementary Fig. 7). Indeed, SnRK3.22 has a full SnRK1 motif in its kinase domain, and SnRK1α1/α2 have a general SnRK phosphorylation motif in the domain of interaction with PP2Cs, suggesting that some SnRK3s may act as SnRK1 upstream kinases, mirroring findings in mammals where AMPK is mainly regulated through phosphorylation by calcium-activated upstream kinases[8]. Genetic analysis showed that *snrk1a2 snrk3.22* has a similar phenotype as the single mutant parents, suggesting they work in the same pathway under salt stress. Thus, SnRK1–SnRK3.15–SnRK3.22 may act as signal integrating points in the salt stress response to promote plant survival under stress (Supplementary Fig. 7).

The ability of cells to respond and adapt to cellular stress is critical for organismal function and development, and this is especially important in sessile organisms like plants. Plant responses to stress appear to have evolved from nutrient and energy sensing, which are activated in the cell in response to most stresses[55]. Currently, the mechanisms underlying SnRK1 role as a master regulator of cellular stress response are poorly understood. The ABA library is enriched for proteins containing a full SnRK1 phosphorylation site, suggesting that SnRK1 may modulate the ABA response by phosphorylating select signaling components. In the future, the identification of SnRK1 kinase substrates among the SnIPs will aid in further elucidating the mechanism of SnRK1 modulation of ABA signaling and stress responses, which is likely to be of great biotechnological importance.

## Methods

**Yeast two-hybrid screen and validation assays**. High-throughput Y2H inter-action assays were conducted by mating using the LexA system, as previously described in Lumba et al.[5]. The six SnRK1 subunits (α1, α2, β1, β2, β3, and βγ) autoactivated the leucine (*LEU*) and *LacZ* reporters when fused to the LexA DNA binding domain (DBD). Therefore, they were fused to the B42 activation domain (AD; pEZY45, high-copy-number plasmid) under the inducible Gal1 promoter and transformed into *Saccharomyces cerevisiae* strain EGY48 (MATα) carrying the chromosomal *LEU* reporter. Cloning primers are listed in Supplementary Table 6. A library of 258 ABA-responsive proteins fused to the LexA DBD (pEZY202, high-copy-number plasmid) and expressed under the constitutive Adh1 promoter was previously transformed in the yeast strain RFY206 (MATa) carrying the *LacZ* reporter (pSH18-34 plasmid). Library autoactivators of both reporters were pre-viously determined as those activating both reporters with >50% of the library clones in Lumba et al.[5]; they are listed in Supplementary Data 1 and were excluded from the analysis. Library clones that autoactivated only one reporter (7 for the *LacZ* and 12 for the *LEU*) are annotated as AA for that reporter and only data for the reporter that did not show autoactivation was used for further analysis (Sup-plementary Data 1, 2, 4). Yeast strains were independently mated (two independent mating reactions per reporter) and interactors were selected on media lacking leucine or supplemented with X-gal, with each plate containing ten empty pray-vector (EV) colonies as the negative controls (Supplementary Fig. 1). Assays were performed in duplicates (two plates per reporter), as described in Lumba et al.[5]. The SnRK1 baits mated with the empty pray vectors did not show autoactivation of the reporters nor caused growth inhibition; the latter is likely due to the inducible Gal1 promoter used to drive expression of the baits. Growth of yeast colonies on solid media supplemented with X-gal was scored visually for the presence of blue color that was darker than the EV colonies, which were mostly white or very light blue. Growth of yeast colonies on solid media lacking leucine was scored using ImageJ[65]; colony size was measured and compared to the average size of all EV colonies on the same plate. Colonies larger than the average size of the EV colonies, at the 99.5% CI threshold, were considered as positive interactions. The results from the Y2H retests using the PRS, HCI, and RRS were scored in a similar manner, but the plates contained six EV colonies.

The positive reference set (PRS) used to determine assay sensitivity included all possible nine literature curated interactions between SnRK1 subunits and library clones present in the library (Supplementary Table 1). Since ABF3, AHG3, and SnRK3.15 autoactivate the *LacZ* reporter[5], they were only tested with the *LEU* reporter. Also, ABI1 autoactivated both reporters[5], thus could not be tested neither in the primary nor in the retest screen. Unfortunately, a true negative reference set could not be assembled, as there is no literature data on negative interactions between SnRK1 and proteins in our screen. Instead, we used a random number

generator to construct a random reference set (RRS) of SnRK1–SnIP pairs that showed no interaction in the initial screen, as in previous studies (Supplementary Table 2[23,25]). All of the library proteins included in the RRS did however interact with other library proteins in the Lumba et al.[5] screen and thus appear to be properly expressed in yeast. In the Y2H retest, the 132 HCI that were selected included positive interactions between the SnRK1α and SnRK1β/βγ subunits with 48 core SnIPs (Fig. 1b and Supplementary Data 4). Interactions that could be tested with only one reporter, due to autoactivation of the other reporter, were excluded.

**Bioinformatics, network analysis, and computational validation**. Previously published protein-protein interactions were obtained from the Arabidopsis Inter-action Viewer (AIV; http://bar.utoronto.ca/interactions). Network interaction maps were created using Cytoscape 3[66]. GO Enrichment analysis was done using the BinGO3 plugin in Cytoscape 3, using the whole Arabidopsis genome as the background and hypergeometric test and Benjamini and Hochberg False Discovery Rate correction for multiple testing correction[67]. GO molecular function annotation was obtained from Gene Ontology Consortium (2016). Subcellular localization data was obtained from SUBACon[68]. PhosMoS (https://doi.org/10.5281/zenodo.3661487) was written with Python2 language using several BioPython packages[69].

Publicly available gene expression data from AtGeneExpress (Bio-Analytic Resource, BAR http://bar.utoronto.ca[70,71]) was used to create heatmap representation of gene expression data using Morpheus (https://software.broadinstitute.org/morpheus). Time course data was used for all time points available (3–4 timepoints), and data was split by tissue (roots and shoots). Correlation in coexpression was calculated using Pearson correlation coefficient (PCC) with a cutoff of 0.75 ($p < 0.01$;[41] for each protein pair over each abiotic stress used, by tissue type. The Clustermaker plugin app[72] was used to cluster all interactions into neighborhoods using the Community clustering (GLay) algorithm[73], in Cytoscape.

In order to see if SnRK1 subunit interactors as measured by Y2H could constitute "real" interactions, we performed co-subcellular localization enrichment analyses, under the assumption that if two proteins interact with each other it is highly likely that they would share the same subcellular localization. The ABA responsive library of 258 proteins[5] was used as the background. For this test, the six subunits were analyzed separately because they are variously located in different organelles. To retrieve the cellular localization information of these six subunits, the "ATH_GO_GOSLIM.txt" file from TAIR was downloaded (the file was dated 2019-01-05), and a "winner-takes-all" strategy was used to flag a protein's localization from the GO CC (cellular component) aspect of this file if there was more than one subcellular compartment associated with a protein. Take the SnRK1βγ, which is located in the nucleus, as an example. In our Y2H experiments, 56 proteins (out of 258 proteins) exhibited positive binding signals with this subunit. Among the 258 proteins, 85 of them are found in the nucleus, allowing us to calculate the expected number of Y2H positive proteins located in the nucleus as $56 \times (85/258) = 18.4$. However, the real number of Y2H positive proteins located in the nucleus is actually 30, indicating an enrichment. A hypergeometric test was used to calculate the statistical significance of this enrichment—the $p$-value is indicated in the Supplementary Fig. 2. We also performed a co-molecular function enrichment analysis for the SnRK1 subunit interactors in a similar manner, except we used the GO MF (molecular function) aspect from the GO Slim file to perform the enrichment tests (Supplementary Fig. 2).

**Bimolecular fluorescence complementation assays**. Predicted SnRK1 inter-actors were tested in *Nicotiana benthamiana* by Bimolecular Fluorescence Complementation assays using a split yellow fluorescent protein (YFP) fusion Gateway vector[5]. Localization constructs were cloned in pEARLEYGATE104 (YFP) vectors. The constructs were transformed into *Agrobacterium tumefaciens* strain GV2260. The transient BiFC assay was done using 4-week-old *Nicotiana benthamiana* plants grown under long days (16 h of light at 28 °C and 8 h of darkness at 24 °C), whose leaves were syringe-infiltrated with *Agrobacterium* at a final OD of 0.2[74]. Fluor-escence was visualized after 72 h using an LSM550 confocal microscope (Zeiss). Confocal images were taken with a 488 nm excitation laser and a 515–535 nm bandpass filter for YFP emission. Plant tissue was directly mounted on glass slides in double distilled water.

**Arabidopsis thaliana mutants and stress assays**. All *Arabidopsis thaliana* mutants used in this study were previously characterized and are all in Col-0 wildtype background: *snrk1a2* (WiscDSLox320B03[75,76]) and *snrk1a1* (GABI_579E09[15,75,76]); *snrk3.22* (SALK_118231C[5]); *duf828/fl7* (SALK_077717[60]) and *myb77-2* (SALK_055373C[61]); *afp1-4 afp2-1*[45]. Gene accession numbers can be found in Supplementary Data 1. All mutants were confirmed to be homozygous for the T-DNA insertion by PCR. For generation of double mutants (*snrka2 snrk3.22* and *snrk1a1 snrk1a2*), reciprocal crosses were conducted, and four siliques were analyzed per cross. No *snrk1a1 snrk1a2* homozygous seedlings were recovered in several generations tested by PCR.

For stress assays, seeds were collected from *Arabidopsis thaliana* (Colombia) plants grown in climate chambers (Enconair) under long days (16 h of light at 22 °C and 8 h of darkness at 18 °C) with white light (100–150 μmol m$^{-2}$ s$^{-1}$). Seeds (2–4 months old) were sterilized, chilled at 4 °C for 4 days and plated as previously

described[75] on ½ strength Murashige and Skoog (MS) medium. For the salt treatment, seeds were plated on 1/2 strength MS medium with or without 100 mM NaCl, or 200 mM/400 mM sorbitol for osmotic treatment. Seedling growth was scored 10 days after imbibition (10 DAI) as the emergence and growth of true leaves; 46–52 seeds were used for each genotype (Supplementary Data 7). The experiments were replicated three to six times, each time using seeds collected from different harvests. The average of biological replicates ± SEM is shown in the graphs. Variance within treatment was calculated using one-way ANOVA at 95% confidence interval. Statistical significance compared to wildtype was calculated using post hoc Tukey HSD test (Supplementary Data 7).

**Statistics and reproducibility**. For the Y2H and validation screens, two separate mating reactions and two distinct reporters were used in duplicates (two plates per reporter). For BiFC assays, three leaves were infiltrated and protein-protein interaction was scored in approximately 100 cells. The assay was repeated three separate times with the same interaction results. For the seedling establishment and growth assays, seeds from three (Fig. 8) or six (Fig. 6) plant harvests were used. A sample of 46–52 seeds from each harvest was assayed for seedling establishment and growth phenotypes. Each batch of seeds is considered a biological replicate. Different harvests were used to ensure reproducibility of the results. One way ANOVA at 95% CI was used to assess variance within seedling establishment/growth and post hoc Tukey HSD test to determine significance difference between the genotypes used, within the same treatment.

**Reporting summary**. Further information on research design is available in the Nature Research Reporting Summary linked to this article.

## Data availability
All relevant data are available in the paper and Supplementary information files. SnRK1 constructs are available from the corresponding author upon request (gazzarrini@utsc.utoronto.ca). Protein–protein interaction data are available at the BAR (www.bar.utoronto.ca), in the Arabidopsis Interactions Viewer 2 and in ePlant (https://doi.org/10.1038/s42003-020-0866-8).

## Code availability
Code for PhosMoS can be downloaded from https://doi.org/10.5281/zenodo.3661487.

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

## Acknowledgements

We thank P. McCourt (University of Toronto) and D. Desveaux (University of Toronto) for supplying Y2H/BiFC vectors and yeast strains. We thank R. Finkelstein (University of California Santa Barbara) for *afp1-4/afp2-1* seeds and A. Mott (University of Toronto Scarborough) for critical reading of the manuscript. This work was supported by an NSERC-DG (Natural Sciences and Engineering Research Council of Canada, Discovery Grant) grant to S.G. and OGS (Ontario Graduate Scholarship) to C.C.

## Author contributions

S.G. and C.C. designed the experiments and wrote the manuscript. C.C. conducted the bioinformatic and network analyses and most of the experiments. A.C. cloned SnRK1b subunits. S.L. trained C.C. in high-throughput Y2H screening. S.D. and N.P. conducted the computational validation. All authors read and approved the manuscript.

## Competing interests

The authors declare no competing interests.
