## [Peer Review File · Communications Biology]

Reviewers' comments:

Reviewer #1 (Remarks to the Author):

Carianopol et al Constructed a detailed protein interaction network for the SnRK1 kinase complex. SnRK1 has well-supported roles in energy metabolism and stress responses, but the targets and protein associated with SnRK1 are still largely unknown. The six subunits of SnRK1 were screened against a library of ABA associated genes and a subset were tested for interactions in planta using BiFC. This approach identified 119 ABA network proteins with uncharacterized interactions with SnRK1. Several putative SnRK1 interacting proteins were functionally characterized and knockouts were found to have altered tolerance to salt, supporting their roles in osmotic stress pathways. Taken together, Carianopol et al. provide the strongest and most compressive evidence to date on SnRK1's role in stress pathways. These results are technically sound and interesting, and I have a few minor comments or concerns that I feel should be addresses to strengthen the manuscript.

It is unclear what backgrounds were used for GO term-based analyses. The 125 high-quality SnIPs show enrichment in functions related to ABA signaling, which is expected as most of the proteins that were screened for interactions have well defined roles in ABA mediated processes (and thus GO terms involved in ABA).

The expression data used to construct Figure 4 is quite old Microarray data, and higher quality, higher resolution RNAseq data exists for various Arabidopsis stresses. Reanalysis of new RNAseq expression data would likely not change the results too significantly, but it would be useful. This may also be beyond the scope of this manuscript.

Broadly, there are a lot of interaction networks in the paper, that are quite similar and include the same nodes with different related datasets. It is also difficult to draw meaning from most of the interaction networks and it is hard to read the names of many individual nodes. I suggest the authors consider if one or more of these should be remade for clarity or moved to the supplement. Other comments are below:

Figure 1 is difficult to read. Because nodes are colored based on function, the names of individual genes could probably be removed, as these are illegible anyway. For comparison, Figure 3A is much easier to read!

Similarly, Figure 2 is also really difficult to read and it's hard to draw meaning out of this cystoscope network. Supplemental Tables S2 and S4 showing GO enrichment are probably sufficient instead of this figure.

Minor:

The merged pdf is > 30,000 pages because of a formatting error that should be corrected.

I noticed a few minor typos and grammatical errors throughout the paper, and the authors should thoroughly edit the text during revision.

Reviewer #2 (Remarks to the Author):

This paper reports a screen for SnRK1 interactors using a library of ABA interacting proteins. It also includes a number of gene set enrichment, expression correlation and network analyses using

previously published data. The results are significant, expanding the putative SnRK1 interaction network significantly over the previously reported results.

Major observations:

1. The screen design contains only duplicates, although triplicates would have been preferred, since using two reporters apparently cannot be used to get more confidence in the hits.
2. The screen uses only a library of selected ABA interacting proteins identified in a previous screen, eliminating non-ABA kinases and other putative SnRK1 interacting proteins, which limits the scope of the work.
3. Additional negative controls would have been useful for increasing confidence in the predictions.
4. Over-representation results may not be that important from characterizing SnRK1 interaction space given the bias in selecting the screen library. The integration of screen results with cellular localization and expression correlation analysis is a plus, adding some confidence in the predicted networks.
5. The selection of the 48 SnRK1 substrates is not well explained, with only one reference citing an instance of SnRK1 subunits interactions (and the reference is missing).
6. Phenotypic analyses: salt stress is usually tested using 250 – 300 mM NaCl; the authors should justify the conditions used for testing salinity tolerance and comment on the interpretation/significance of the results obtained with 100 mM NaCl

Minor technical points:

1. Although the references list is extensive, some are missing (i.e. (Polge et al., 2008)) probably because an automated reference editing tool was not used.
2. Also, it was difficult to obtain the table data reported in the paper. Probably it was miss-formatted during the submission. It would be preferable to deposit the raw data in a public database for access.

Reviewer #3 (Remarks to the Author):

The manuscript describes the mapping and analysis of the interactome network between regulatory and catalytic subunits of the plant SnRK family of protein kinases and proteins whose encoding transcripts are upregulated following ABA treatment. After initial mapping, the authors aim to identify and analyse putative kinase substrates, describe potential roles of the interactors in plant stress responses and finally conduct some in vivo validations of the interactions and of selected identified interactors in stress responses.

The topic is very exciting and highly important. The connection between ABA signaling and the central kinase regulating the response of metabolic activities is as underexplored as it is important in light of climate change and the consequences for crops that are predicted in its wake. Similarly, the chosen systematic approach needs to be applauded as systematic studies enable discovery of systems level and unexpected relationships that classic hypothesis driven approaches are blind towards. Lastly, the dataset will undoubtedly be an important resource that will inspire many follow-up studies.

At the same time, the paper suffers from a few shortcomings, which may in part due to missing details, that need to be addressed. The most pervasive appears to be a potentially circular logic in some of the bioinformatic analysis as described below.

Interactome mapping:

1. The Lumba lab has a strong record in interactome analysis of the ABA signaling system and the data are generally trustworthy, especially after the filtering. I would suggest using a medium-scale set of at least 50 well documented interactions and 50 random protein pairs as negative control to benchmark their screening system. This would enable to measure experimentally measuring the

precision and background of requiring both or a single reporter for scoring yeast growth as evidence for a protein interaction. The authors should also comment on how the identification of auto-activators has been conducted.

1b. While this is to some extent a matter of personal preference, I am wondering why all primary hits are labeled 'interactors'. As in any system, Y2H may give rise to artifacts and in no assay should all primary hits be labeled interactors. I would suggest to reserve this meaningful attribute to what they label high-confidence interactors.

2. In the analysis in lines 127 and 129 it is unclear what reference sample was used to measure enrichment, what test was used and how the multiple hypothesis correction was done. As a reference, the screened set of 258 screened interactors should be used. If a Fisher has not sufficient power to detect significance, especially after correcting for multiple hypotheses, perhaps a degree-preserving random rewiring approach may yield a signal. The issue of missing indication of the reference space for enrichment analyses is a common issue in the manuscript.

3. Line 151: please indicate that this is a set of predicted interactions. Also, please justify why predicted instead of experimentally demonstrated interactions were chosen for this analysis.

4. Ln 157: Are the numbers significant, or do they correspond to the relative proportions in the set of 258 input genes from which all interactors were sampled?

5. Ln 160: as before, please provide base distribution for the analysis, statistical test and method for multiple hypothesis correction

6. Ln 166: as #6

7. The rationale for the substrate analysis is plausible, however given that the potential recognition sequences are present in a larger proportion of all SnIP than the putative substrates, it seems the first selection criteria (interacting with reg. and cat. subunit) did not work. The authors should discuss this discrepancy and reconsider their identification criteria. Perhaps, in vitro kinase assays for small sample from both subsets can clarify which criteria (interactions vs motif) are more reliable. Such assays would strongly support the entire analysis.

In light of the skepticism I have towards the 48 network-based substrates, I am very skeptical of the following analyses done with this subset as 'substrates'. Additionally, the conclusion from this analysis does not seem overwhelming.

8. Ln 220: aren't these substrates already highly connected by virtue of their selection as proteins interacting with a regulatory and catalytic subunit?

9. The results of the in planta BiFC validations are really nice.

10. Ln 316: which protein interaction data were integrated

11. Ln 323: As the network was generated by starting from the two kinases, it is by nature of this analysis that SnRK1 and 3 are articulation points and is not a property of the (biological network). Please be careful not to confuse an effect (e.g. of node removal) on the graph in cytoscape, with an effect on the in vivo network that may result in phenotypes. Please rethink this analysis and the wording.

Minor issues:

12. Ln 71: Given the essential and central nature of metabolism I am not sure why a role in metabolic control is insufficient to explain the lethal phenotype. Perhaps consider rephrasing this sentence.

13. Ln 225/226: please check this sentence for language.

14. Ln 269: going back to my point #1: if the authors believe their filtering removes false positives, i.e. artifacts of their experimental assay, then all data prior to this filtering could be termed Y2H candidate pairs or some other name to distinguish what the authors consider bona fide Y2H interactions from artifacts (which every assay has).

15. Ln 298: given that not all interactions were validated, perhaps rephrase as "Thus, several exemplary SnRK1-..."

16. Ln 310: I think a word is missing in this sentence.

17. Ln 348: are these ALL known pairs that could be detected?

Reviewers' comments:

Reviewer #1 (Remarks to the Author):

Carianopol et al Constructed a detailed protein interaction network for the SnRK1 kinase complex. SnRK1 has well-supported roles in energy metabolism and stress responses, but the targets and protein associated with SnRK1 are still largely unknown. The six subunits of SnRK1 were screened against a library of ABA associated genes and a subset were tested for interactions in planta using BiFC. This approach identified 119 ABA network proteins with uncharacterized interactions with SnRK1. Several putative SnRK1 interacting proteins were functionally characterized and knockouts were found to have altered tolerance to salt, supporting their roles in osmotic stress pathways. Taken together, Carianopol et al. provide the strongest and most compressive evidence to date on SnRK1's role in stress pathways. These results are technically sound and interesting, and I have a few minor comments or concerns that I feel should be addresses to strengthen the manuscript.

It is unclear what backgrounds were used for GO term-based analyses. The 125 high-quality SnIPs show enrichment in functions related to ABA signaling, which is expected as most of the proteins that were screened for interactions have well defined roles in ABA mediated processes (and thus GO terms involved in ABA).

> The background for the GO term based analyses used is the Arabidopsis genome. We agree with the reviewer that since we screened an ABA-responsive library, the enrichment of proteins with functions in ABA-mediated processes is expected. We have updated the Results section (lines 202 and 207).

The expression data used to construct Figure 4 is quite old Microarray data, and higher quality, higher resolution RNAseq data exists for various Arabidopsis stresses. Reanalysis of new RNAseq expression data would likely not change the results too significantly, but it would be useful. This may also be beyond the scope of this manuscript.

> Although high resolution RNAseq data now exists for various stresses, we chose to use the microarray data from Kilian et al., (2007) since the data concerning various stresses was all collected from one experiment and thus expression change between stresses can be comparable. However, we agree with the reviewer that it would be interesting to see if similar results could be obtained by using RNAseq data as well, although that is beyond the scope of this manuscript since our data analysis using the microarray data has yielded hypothesis which were later supported by further functional studies.

Broadly, there are a lot of interaction networks in the paper, that are quite similar and include the same nodes with different related datasets. It is also difficult to draw meaning from most of the interaction networks and it is hard to read the names of many individual nodes. I suggest the authors consider if one or more of these should be remade for clarity or moved to the supplement. Other comments are below:

> We believe the bioinformatic and network analysis is important for conclusions drawn in the paper. However, we have removed Fig 1c (of the previous version), which showed the extended SnRK1-SnIPs network and included interactions between all proteins using data from the Arabidopsis Interaction Viewer), because part of Figure 7a includes the extended interaction network.

Figure 1 is difficult to read. Because nodes are colored based on function, the names of individual genes could probably be removed, as these are illegible anyway. For comparison, Figure 3A is much easier to read! Similarly, Figure 2 is also really difficult to read and it's hard to draw meaning out of this cystoscope network. Supplemental Tables S2 and S4 showing GO enrichment are probably sufficient instead of this figure.

> We agree. We have removed the node labels (gene names) from Figure 1 and modified some figures to make them easier to read (see also below). We also removed Fig 2 of the previous version (Functional analysis of ABA-responsive SnRK1 interacting proteins) since the enrichment data is shown in Supplementary Data 5 of the new version. All Figure numbers after Figure 1 have thus been updated.

Minor:

The merged pdf is > 30,000 pages because of a formatting error that should be corrected.

> Yes, there was a formatting error when the files were converted from Excel to PDF. Tables should be in the correct format in the resubmitted manuscript. They should be in the right format this time.

I noticed a few minor typos and grammatical errors throughout the paper, and the authors should thoroughly edit the text during revision.

> We have reviewed and edited any errors we found throughout the paper.

Reviewer #2 (Remarks to the Author):

This paper reports a screen for SnRK1 interactors using a library of ABA interacting proteins. It also includes a number of gene set enrichment, expression correlation and network analyses using previously published data. The results are significant, expanding the putative SnRK1 interaction network significantly over the previously reported results.

Major observations:

1. The screen design contains only duplicates, although triplicates would have been preferred, since using two reporters apparently cannot be used to get more confidence in the hits.

> We agree that triplicates would have resulted in higher confidence interactions. Nevertheless, we used two reporters in duplicates, for a total of 4 interaction screens per

bait (yeast strains were mated independently in each replica). Additionally, as per reviewer 3 comments, we have determined the quality of our initial Y2H screen by determining assay sensitivity, background and retesting of over 100 interactions using both reporter system (lines 124-160).

2. The screen uses only a library of selected ABA interacting proteins identified in a previous screen, eliminating non-ABA kinases and other putative SnRK1 interacting proteins, which limits the scope of the work.

> We believe our screen was rather targeted then limited. Our aim was to understand to what extent SnRK1, which is a major and conserved regulator of metabolism under conditions of stress in eukaryotes, interacts with signaling components of the well-known stress hormone ABA. This stems from a previous study (Rodrigues et al., 2013), which used microarray data to show an overlap in downstream components of the ABA and SnRK1 response pathways.

3. Additional negative controls would have been useful for increasing confidence in the predictions.

> In the initial Y2H screen, each plate had ten colonies harbouring the empty vectors as negative controls. Furthermore, the library was previously screened and clones that autoactivated both reporters with > than 50% of the library clones had been identified and removed from further analysis (Lumba et al., 2014). We have updated the Methods section to contain more details about the empty vector controls and autoactivators (lines 446-450). As mentioned above, we have now determined assay background, sensitivity and retest of >200 interactions (lines 124-160).

4. Over-representation results may not be that important from characterizing SnRK1 interaction space given the bias in selecting the screen library. The integration of screen results with cellular localization and expression correlation analysis is a plus, adding some confidence in the predicted networks.

> We agree with the reviewer that since we screened an ABA-responsive library, the enrichment of proteins with functions in ABA-mediated processes is expected. We have specified this in text (lines 207).

5. The selection of the 48 SnRK1 substrates is not well explained, with only one reference citing an instance of SnRK1 subunits interactions (and the reference is missing).

> We have reworded the selection criteria to more accurately describe the set of 48 core SnIPs (lines 148-152). More references have been included and the reference list has been updated. In light of the comments from Reviewer 3, who pointed out that the 48 subs are not enriched for SnRK1 phosphorylation sites, we renamed the set of 48 SnIPs as ‘core SnIPs’.

6. Phenotypic analyses: salt stress is usually tested using 250 – 300 mM NaCl; the authors should justify the conditions used for testing salinity tolerance and comment on the interpretation/significance of the results obtained with 100 mM NaCl

> We have chosen 100mM NaCl because this concentration reduces growth of Arabidopsis WT seedlings and inhibits that of well-known salt hypersensitive mutants, such as *sos1-3* (Liu and Zhu 1998, Science; Zhu et al., 1998, Plant Cell). Therefore, 100mM is considered a salt stress condition for Arabidopsis growth (Song et al., 2017; Zhang et al., 2017). Furthermore, we used a concentration of salt that would not completely inhibit growth of WT Arabidopsis seedlings to allow the identification of both positive and negative regulators of the salt stress response. Although WT Arabidopsis seedlings germinate at 200mM and higher concentrations, they die before the emergence of true leaves (which we assayed as seedling growth in our experiments).

Minor technical points:

1. Although the references list is extensive, some are missing (i.e. (Polge et al., 2008)) probably because an automated reference editing tool was not used.

> We have updated the reference list.

2. Also, it was difficult to obtain the table data reported in the paper. Probably it was miss-formatted during the submission. It would be preferable to deposit the raw data in a public database for access.

Yes, there was a formatting error when the files were converted from xls to PDF. We are sorry about this inconvenience. Tables should be in the correct format in the resubmitted manuscript. The data will be deposited in the Arabidopsis Interaction Viewer.

Reviewer #3 (Remarks to the Author):

The manuscript describes the mapping and analysis of the interactome network between regulatory and catalytic subunits of the plant SnRK family of protein kinases and proteins whose encoding transcripts are upregulated following ABA treatment. After initial mapping, the authors aim to identify and analyse putative kinase substrates, describe potential roles of the interactors in plant stress responses and finally conduct some in vivo validations of the interactions and of selected identified interactors in stress responses.

The topic is very exciting and highly important. The connection between ABA signaling and the central kinase regulating the response of metabolic activities is as underexplored as it is important in light of climate change and the consequences for crops that are predicted in its wake. Similarly, the chosen systematic approach needs to be applauded as systematic studies enable discovery of systems level and unexpected relationships that classic hypothesis driven approaches are blind towards. Lastly, the dataset will undoubtedly be an important resource that will inspire many follow-up studies.

At the same time, the paper suffers from a few shortcomings, which may in part due to missing details, that need to be addressed. The most pervasive appears to be a potentially circular logic in some of the bioinformatic analysis as described below.

Interactome mapping:

1. The Lumba lab has a strong record in interactome analysis of the ABA signaling system and the data are generally trustworthy, especially after the filtering. I would suggest using a medium-scale set of at least 50 well documented interactions and 50 random protein pairs as negative control to benchmark their screening system. This would enable to measure experimentally measuring the precision and background of requiring both or a single reporter for scoring yeast growth as evidence for a protein interaction. The authors should also comment on how the identification of auto-activators has been conducted.

> We have determined the quality of our initial Y2H screen by retesting > 200 interactions using a PRS and RRS, as suggested (lines 124-147). Unfortunately, only 9 literature curated interactions could be tested (SnRK1 interactors present in the library), therefore we have included analysis of 15 literature curated interactions from Lumba et al. 2014. Furthermore, we retested >100 HCI identified after filtering (lines 148-160). Overall, the confirmation rates of PRS and HCI with *LacZ* and *LEU* reporters range from 50-67%, suggesting that both reporters are appropriate for identifying novel PPIs in this system. We thank the reviewer for this suggestion, which improved the manuscript.

> Library autoactivators of both reporters were previously determined as those activating both reporters with > than 50% of the library clones in Lumba et al. (2014); they are listed in Supplementary Data 1 and were excluded from the analysis. Library clones that autoactivated only one reporter (7 for the *LacZ* and 12 for the *LEU*) are annotated as AA for that reporter and only data for the reporter that did not show autoactivation was used for further analysis (Supplementary Data 1-3). This is included in the Methods section (lines 456-450).

1b. While this is to some extent a matter of personal preference, I am wondering why all primary hits are labeled 'interactors'. As in any system, Y2H may give rise to artifacts and in no assay should all primary hits be labeled interactors. I would suggest to reserve this meaningful attribute to what they label high-confidence interactors.

> We have edited the text (lines 104, 110-111).

2. In the analysis in lines 127 and 129 it is unclear what reference sample was used to measure enrichment, what test was used and how the multiple hypothesis correction was done. As a reference, the screened set of 258 screened interactors should be used. If a Fisher has not sufficient power to detect significance, especially after correcting for multiple hypotheses, perhaps a degree-preserving random rewiring approach may yield a signal. The issue of missing indication of the reference space for enrichment analyses is a common issue in the manuscript.

> The reference set used to measure enrichment in Supplementary figure 2 was the set of 258, which was screened for interactors (line 498). A hypergeometric test was used for

statistical significance (line 508-510). We have updated any enrichment statements with the background they are referring to (lines 171; 207)

3. Line 151: please indicate that this is a set of predicted interactions. Also, please justify why predicted instead of experimentally demonstrated interactions were chosen for this analysis.

> As suggested by Reviewer 1, we have removed the previous Fig 1c and related text, which this sentence refers to.

4. Ln 157: Are the numbers significant, or do they correspond to the relative proportions in the set of 258 input genes from which all interactors were sampled?

> All proportions correspond to those found in the library, unless specified otherwise, and we have added this to the text (lines 167).

5. Ln 160: as before, please provide base distribution for the analysis, statistical test and method for multiple hypothesis correction

> The statistical analysis was done within Bingo by Cytoscape. The app uses Hypergeometric test for statistical significance and Benjamini and Hochberg False Discovery Rate correction for multiple testing correction. The reference set is the Arabidopsis genome. This was updated within the methods section (line 498).

6. Ln 166: as #6

7. The rationale for the substrate analysis is plausible, however given that the potential recognition sequences are present in a larger proportion of all SnIP than the putative substrates, it seems the first selection criteria (interacting with reg. and cat. subunit) did not work. The authors should discuss this discrepancy and reconsider their identification criteria. Perhaps, in vitro kinase assays for small sample from both subsets can clarify which criteria (interactions vs motif) are more reliable. Such assays would strongly support the entire analysis.

In light of the skepticism I have towards the 48 network-based substrates, I am very skeptical of the following analyses done with this subset as 'substrates'. Additionally, the conclusion from this analysis does not seem overwhelming.

> We agree with the reviewer and reorganized this section. We changed the labelling of the 48 potential substrates to "48 core SnIPs" and suggested that some may represent potential substrates based on the presence of SnRK1 consensus motif in (lines 148-153, 182-185). Although phosphorylation assays would aid in determining whether the presence of the phosphorylation motif is important in identifying substrates, this is beyond the scope of this study. Furthermore, these assays aren't easily to performed in a short time-frame, especially for SnRK1 which works in a complex; each protein needs to be expressed and purified from *E. coli*. We are currently troubleshooting a radioactive-free method (Phos-tag) that can be used to test several potential substrates. We hope we will be able to include the results in a follow-up manuscript, which will have a focus on phosphorylation targets of SnRK1.

8. Ln 220: aren't these substrates already highly connected by virtue of their selection as proteins interacting with a regulatory and catalytic subunit?

> The network of 48 SnIPs shown in Figure 2 comprises 314 total interactions and includes 204 SnRK1-SnIPs, but also 110 interactions amongst the SnIPs themselves (SnIP-SnIP) obtained from published data sets (lines 254-255). This would not necessarily result in a highly connected network, as the SnIPs would not necessarily need to interact with one another, outside of the SnRK1 subunits.

9. The results of the in planta BiFC validations are really nice.

> Thank you.

10. Ln 316: which protein interaction data were integrated

> Protein-Protein interaction data from the Arabidopsis Interaction and our SnRK1 complex interactions. We have clarified this in the text (lines 253-246; 498).

11. Ln 323: As the network was generated by starting from the two kinases, it is by nature of this analysis that SnRK1 and 3 are articulation points and is not a property of the (biological network). Please be careful not to confuse an effect (e.g. of node removal) on the graph in cytoscape, with an effect on the in vivo network that may result in phenotypes. Please rethink this analysis and the wording.

> We appreciate the reviewer's comment and have removed the sentence about articulation nodes.

Minor issues:

12. Ln 71: Given the essential and central nature of metabolism I am not sure why a role in metabolic control is insufficient to explain the lethal phenotype. Perhaps consider rephrasing this sentence.

> We have updated the text and this sentence now reads "supporting SnRK1 function as metabolic regulator and suggesting additional roles in development" (line 71-72).

13. Ln 225/226: please check this sentence for language.

14. Ln 269: going back to my point #1: if the authors believe their filtering removes false positives, i.e. artifacts of their experimental assay, then all data prior to this filtering could be termed Y2H candidate pairs or some other name to distinguish what the authors consider bona fide Y2H interactions from artifacts (which every assay has).

> We have updated the text relative to ANAC55, which was deemed a false positive in the primary screen according to our filtering criteria (lines 288).

15. Ln 298: given that not all interactions were validated, perhaps rephrase as "Thus, several exemplary SnRK1-..."

> We have modified this in text, which now reads: “Overall, six of the eight (75%) positive Y2H interactions were validated *in planta*, a rate similar to those obtained for PRS and HCI validation assays (Fig 1c). This is in agreement with previous high-throughput verifications rates using an independent assay (Rual et al., 2005; Lumba et al., 2014) and confirms that our filtered and curated Y2H interactions are of high quality. (lines 300-303)

16. Ln 310: I think a word is missing in this sentence.

>Fixed (now line 337)

17. Ln 348: are these ALL known pairs that could be detected?

>Yes, in the whole screen, there are a total of 9 previously published interactions, 6 of which were confirmed in our primary screen or retest. 8 were confirmed in at least 2 replicates between primary and secondary screen (Lines 125-135; Supplementary Table 1).

REVIEWERS' COMMENTS:

Reviewer #1 (Remarks to the Author):

The authors have addressed my previous concerns

Reviewer #3 (Remarks to the Author):

The authors have thoroughly addressed the concerns and I believe the exciting and important paper can be published as is.